# Atomically dispersed cobalt catalyst anchored on nitrogen-doped carbon nanosheets for lithium-oxygen batteries

Peng Wang[1], Yingying Ren[1], Rutao Wang[1], Peng Zhang[1], Mingjie Ding[1], Caixia Li[1], Danyang Zhao[1], Zhao Qian[1], Zhiwei Zhang[1], Luyuan Zhang[1] & Longwei Yin [1✉]

Developing single-site catalysts featuring maximum atom utilization efficiency is urgently desired to improve oxidation-reduction efficiency and cycling capability of lithium-oxygen batteries. Here, we report a green method to synthesize isolated cobalt atoms embedded ultrathin nitrogen-rich carbon as a dual-catalyst for lithium-oxygen batteries. The achieved electrode with maximized exposed atomic active sites is beneficial for tailoring formation/decomposition mechanisms of uniformly distributed nano-sized lithium peroxide during oxygen reduction/evolution reactions due to abundant cobalt-nitrogen coordinate catalytic sites, thus demonstrating greatly enhanced redox kinetics and efficiently ameliorated over-potentials. Critically, theoretical simulations disclose that rich cobalt-nitrogen moieties as the driving force centers can drastically enhance the intrinsic affinity of intermediate species and thus fundamentally tune the evolution mechanism of the size and distribution of final lithium peroxide. In the lithium-oxygen battery, the electrode affords remarkably decreased charge/discharge polarization (0.40 V) and long-term cyclability (260 cycles at 400 mA g$^{-1}$).

---

[1] Key Laboratory for Liquid-Solid Structural Evolution and Processing of Materials, Ministry of Education, School of Materials Science and Engineering, Shandong University, Jinan 250061, PR China. ✉email: yinlw@sdu.edu.cn

A protic lithium-oxygen (Li-O$_2$) batteries have opened up a new avenue to surmount the flourishing benchmark of lithium-ion counterparts largely in virtue of their unprecedented gravimetric energy density (~3505 Wh kg$^{-1}$, based on $O_2 + 2Li^+ + 2e^- \leftrightarrow Li_2O_2$)[1–4]. However, in terms of cathode, the intractable hurdles regarding inferior redox overpotential and notorious operation stability originated from the insoluble and insulated discharge product of lithium peroxide (Li$_2$O$_2$) overshadow its commercial viability[5–10]. In response, substantial strategies focused on rationally constructing hierarchical porous architectures loaded with various kinds of electrocatalytic nanoparticles (carbonaceous materials[11,12], noble metals[13–15], metal oxides[16–18], sulfides[19,20], etc.) have sprung up to improve oxygen redox kinetics and thus ameliorate over-potentials between oxygen evolution and reduction reactions, referred to OER and ORR, respectively. Nevertheless, in practice, maximizing accessible active sites and enhancing catalytic efficiency have been ignored but a barycenter of strategic importance to essentially enhance redox activity in the field of Li-O$_2$ battery.

Relative to solid nanoparticles, owing to the super-high utilization of active atoms, non-saturated atomic coordination sites and uniformity of active centers, single-atoms (SAs) species as "Holy Grail" deservedly stand out and occupy research frontier in massive catalytic systems embracing water splitting[21–26], fuel cells[27,28], zinc-air batteries[29–32], nitrogen reduction[33], and CO$_2$ reduction reactions[34–39]. Especially, in the fields of rechargeable aprotic battery technologies, atomically dispersed metals as active sites also begin to demonstrate considerable potential in accelerating the conversion kinetics and boosting active species utilization efficiently. For example, when introduced to Li and Na-sulfur batteries, SAs can conspicuously suppress the "shuttle effect" and electrocatalyze the polysulfide reduction[40–42]. Moreover, Gong's group reported that when applied in Li metal anodes, the isolated metal atoms could also affect Li deposition and alleviate dendritic Li growth during the cycles[43]. However, to date, there is almost no report about SA catalysts for ORR/OER in lithium-oxygen batteries. Crucially, the straightforward modulation effect of single atoms on the morphology, distribution and crystalline characteristic of Li$_2$O$_2$ is still in its infancy, which are the key factors governing the reversibility and durability of Li-O$_2$ batteries. On the other hand, metal-organic-frameworks (MOFs) especially zeolitic-imidazolate-framework-8 (ZIF-8) have flourished as sacrificial templates to fabricate atomically dispersed metal-N-C catalysts[27,28,32,35,44,45]. It is worthy pointing out that there are fewer concerns about two-dimensional (2D) MOFs with fruitful metal/nitrogen-containing coordination derivatives to capture single atoms which possess more exposed active sites, large surface/volume ratio and nanoscale thickness. In principle, synergistically coupling single atoms with 2D MOFs-derived carbon matrix as "heaven-made match" can thoroughly unlock its potential in tuning catalytic activity and catalyst utilization efficiency. Whereas synthesis of such splendid catalysts by means of a facile and cost-effective procedure still remains a daunting challenge. Recently, Li and Wu's group reported a pioneering thermal emitting strategy to obtain isolated Cu-SAs/N-C and Pt-SAs/graphene catalysts from bulk metals[46,47]. These studies involve indispensably employing toxic NH$_3$ to coordinate with metal atoms under high temperature. In this regard, it would be quite attractive and urgent to investigate a more rational protocol free of toxic to trap single atoms.

Herein, we put forward a green gas-migration-trapping strategy to synthesize desirable Co single atoms embedded in ultrathin Zn-hexamine complex-derived nitrogen-doped carbon matrix (Co-SAs/N-C) as a dual-catalyst for lithium-oxygen batteries. Benefiting from the advantages of both 2D MOFs and uniformly isolated dispersion of atomic metal sites, the well-defined Co-SAs/N-C catalyst can provide low-impedance charge transfer pathways, and offer abundant specific area for Li$_2$O$_2$ accommodations. More notably, by virtue of density functional theory (DFT) simulations, we conclude that rich Co-N$_4$ moieties functioning as catalytic centers can drastically strengthen the intrinsic lithium superoxide (LiO$_2$)-adsorption ability and thus fundamentally modulate the size, morphology and distribution of involved Li$_2$O$_2$. The electrode design can promote the formation of uniformly distributed nano-sized Li$_2$O$_2$ species during ORR while the decomposition during OER becomes easier as a result of better contact between Li$_2$O$_2$ and catalyst as well as abundant Co-N$_4$ catalytic sites. Accordingly, redox kinetics and ORR/OER overpotentials are efficiently ameliorated. In the Li-O$_2$ battery, the Co-SAs/N-C electrode can afford remarkably decreased charge/discharge polarization (0.40 V versus Li/Li$^+$), superior discharge capacity (20,105 mAh g$^{-1}$ at 200 mA g$^{-1}$, 11,098 mAh g$^{-1}$ at 1 A g$^{-1}$), good cyclability performance (260 cycles at 400 mA g$^{-1}$). This work provides insights into the critical role of Co-N$_4$ species in tailoring the formation routes of Li$_2$O$_2$, and the electrocatalyst design could be extended to other metal-oxygen batteries.

## Results

**Microstructure characterizations.** Supplementary Fig. 1 demonstrates the programmable fabrication process of the Co-SAs/N-C. Firstly, ultrathin N-doped carbon nanosheet (N-C) products are obtained through a fast pyrolysis process of Zn-hexamine white precipitations (as seen in the Method part). The X-ray diffraction (XRD) results demonstrate the successful fabrication of Zn-hexamine samples (Supplementary Fig. 2)[48]. The field-emission scanning electron microscopy (FESEM) images of N-C in Supplementary Fig. 3 illustrate the crumpled ultrathin nanosheet morphology. In the second step, a gas-migration-trapping procedure without toxic NH$_3$ assistance is carried out, in which CoCl$_2$•6H$_2$O is firstly sublimated, then trapped, reduced and stabilized on N-rich supports by the strong interactions between Co and highly electronegative N species according to the thermogravimetric curves in Supplementary Fig. 4. As a contrast, as shown in Supplementary Figs. 5 and 6, the Co nanoparticle trapped N-rich carbon (Co-NPs/N-C) catalysts are also generated just by modulating the annealing temperatures. Then, the corresponding color evolution from white to black of the samples is shown in Supplementary Fig. 7. Figure 1a depicts the FESEM result of Co-SAs/N-C, indicating geometrically 2D nanosheet morphology of the N-C matrix. For Co-SAs/N-C, as displayed in the aberration-corrected high-angle annular dark-field scanning transmission electron microscopy (HAADF-STEM) in Fig. 1b, the transmission electron microscopy (TEM) images of in Fig. 1c together with Supplementary Fig. 8a, b, the almost transparent N-C substrate demonstrates ultrathin graphene-like structure without observed nanoparticles or other impurity species. The HAADF-STEM images with sub-Å resolution in Fig. 1d, e and Supplementary Fig. 9a, b confirm that the uniformly distributed atomically isolated Co atoms (brighter dots) without aggregation can be distinctly observed on the entire N-rich carbon nanosheets. For visual identification, some Co single atoms are highlighted by green circles. Figure 1c (inset) depicts the ring-like selected-area electron diffraction (SAED) pattern of Co-SAs/N-C, indicating a weak crystallinity without diffraction rings distinguished as Co nanoparticle. In sharp contrast, Co-NPs/N-C shows apparent aggregation of Co atoms, indicating the formation of nanoparticles (Supplementary Fig. 10). Furthermore, elemental mapping images in Fig. 1f confirm C, N, and Co elements are homogeneously distributed within the whole carbon scaffold. In addition, inductively coupled plasma optical emission spectroscopic (ICP-OES) testing discloses the mass percent content of single cobalt sites in Co-SAs/N-C sample is about 2.01 wt% (Supplementary Table 1).

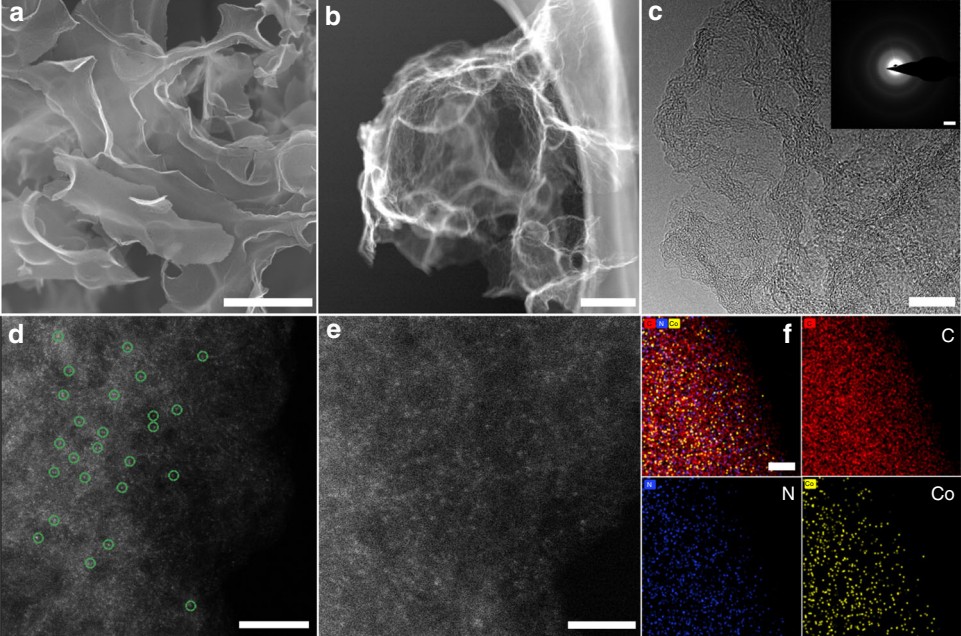

**Fig. 1 Microstructure and morphology characterizations of Co-SA/N-C catalyst. a** SEM image. Scale bar, 1 μm. **b** HAADF-STEM image. Scale bar, 200 nm. **c** HRTEM image. Scale bar, 10 nm. SAED pattern (inset). Scale bar, 2 1/nm. **d, e** Zoom-in HAADF-STEM image. Scale bar, 5 and 2 nm, respectively. **f** Elemental mapping images. Scale bar, 20 nm.

The typical X-ray photoelectron spectroscopy (XPS) spectra of Co-NPs/N-C and Co-SAs/N-C (Supplementary Fig. 11), confirm distinct signals of Co, N, C and O elements. The high-resolution N 1s spectra in Fig. 2a can all be fitted with four peaks, namely, oxidized N, graphitic, pyrrolic and pyridinic groups for the three samples, respectively[48]. Interestingly, for Co-SAs/N-C, the strongest pyridinic N peak displays a ~0.3 eV offset to a larger bonding energy, versus that of Co-NPs/N-C and N-C. Specially, in the high-resolution Co 2p XPS spectra (Fig. 2b), for Co-NPs/N-C, the peaks of Co $2p_{2/3}$ and Co $2p_{1/2}$ are located at 778.5 and 793.9 eV, respectively. Interestingly, for Co-SAs/N-C, the corresponding peaks situated at 780.8 and 795.8 eV, also show a distinct up-shifting tendency. The peak of Co $2p_{2/3}$ for Co-SAs/N-C is centered between $Co^0$ and $Co^{2+}$, illustrating its ionic $Co^{\delta+}$ ($0 < \delta < 2$) characteristic[29,31,35,47,49,50]. It can be inferred that pyridinic N as the predominant configuration can strongly coordinate with single Co atoms to construct $Co-N_x$ structures. Moreover, the X-ray absorption near-edge structure spectra (XANES) of Co K-edge in Fig. 2c demonstrate that both the intensity and position of the fingerprint peak (7729 eV) of Co-SAs/N-C are located in the middle of the standard samples Co foil and CoO, implying that single cobalt atoms in Co-SAs/N-C are being oxidized status, which is in accordance with the XPS results. In addition, Co-SAs/N-C demonstrates a pre-edge peak at about 7711 eV, which is attributed to a fingerprint of $Co-N_4$ square-planar structures[23,30,40]. The Fourier transform $k^3$-weighted extended X-ray absorption fine structure (FT-EXAFS) spectra in Fig. 2d (R space) and Supplementary Fig. 12 (k space) show that only one distinct shell (1.36 Å) corresponding to Co-N scattering path is observable for Co-SAs/N-C, totally different from those of Co-Co bond (2.17 Å) in Co foil and Co-O peak (1.70 Å) in CoO. As depicted in Fig. 2e, for Co-SAs/N-C, the wavelet transform (WT) analysis of EXAFS signal just depicts a maximum located at 4.3 Å$^{-1}$ assigned to Co-N contribution. In sharp contrast, the corresponding maxima located at 6.4 Å$^{-1}$ (Co-Co) and 3.5 Å$^{-1}$ (Co-O) are observed for Co foil and CoO, respectively. The Co K-edge EXAFS fitting of the first shell in R space (Fig. 2f) can be well performed following the Co-N

scattering paths. Based on this route, the fitted curve in k space (Fig. 2g) is also well-matched with the experimental data. Then, to achieve the quantitative bonding information, we extract bond lengths and metal coordination numbers from Co K-edge EXAFS curve fitting. For Co-SAs/N-C, as shown in Supplementary Table 2, the coordination number (N) of the isolated cobalt is 3.8. Together with the pre-edge characteristic peak in Fig. 1c, we infer that the four-coordinated $CoN_4$ configurations are formed in the Co-SAs/N-C species, identical to those reported works[30,40,51,52]. Based on the above results, ultrathin N-doped porous carbon nanosheets anchoring massive homogenously scattered isolated Co atoms are fully certificated. Furthermore, Raman spectra of the three species in Supplementary Fig. 13 reveal that after coupling with single Co atoms, the proportion of the D and G peaks is significantly increased, indicating much local defects are induced in the carbon substrate[23,46,50]. Additionally, Brunauer-Emmett-Teller (BET) analysis (Supplementary Fig. 14) shows that Co-SAs/N-C possesses a considerable specific surface area of 707 m$^2$ g$^{-1}$ with typical type-IV isotherm characteristic.

**Electrocatalytic performance.** To testify the unique electrocatalytic capability of the elaborately designed Co-SAs/N-C species with isolated atom feature toward ORR/OER, a series of electrochemical tests are conducted. Figure 3a exhibits the cyclic voltammograms (CV) profiles of the three samples at a scanning speed of 0.1 mV s$^{-1}$ within 2.0–4.5 V. During the cathodic sweep, the three catalysts show similar reduction peak (2.32 V), corresponding to the ORR process associated with $Li_2O_2$ product ($O_2 + 2Li^+ + 2e^- \rightarrow Li_2O_2$). Meanwhile, during the anodic scan, a clear oxidation peak located at 3.3 V can also be noticeable for the three catalysts, which is associated with the bulk decomposition process for the discharge species ($Li_2O_2 \rightarrow O_2 + 2Li^+ + 2e^-$). However, compared with that of N-C and Co-NPs/N-C counterparts, much higher anodic peak current and large integration area are achieved for the Co-SAs/N-C electrode, implying that more discharge products can be reversibly decomposed and thus much ameliorated catalytic kinetics performance. By contrast, Co-NPs/N-C and N-C-based electrodes experience severer

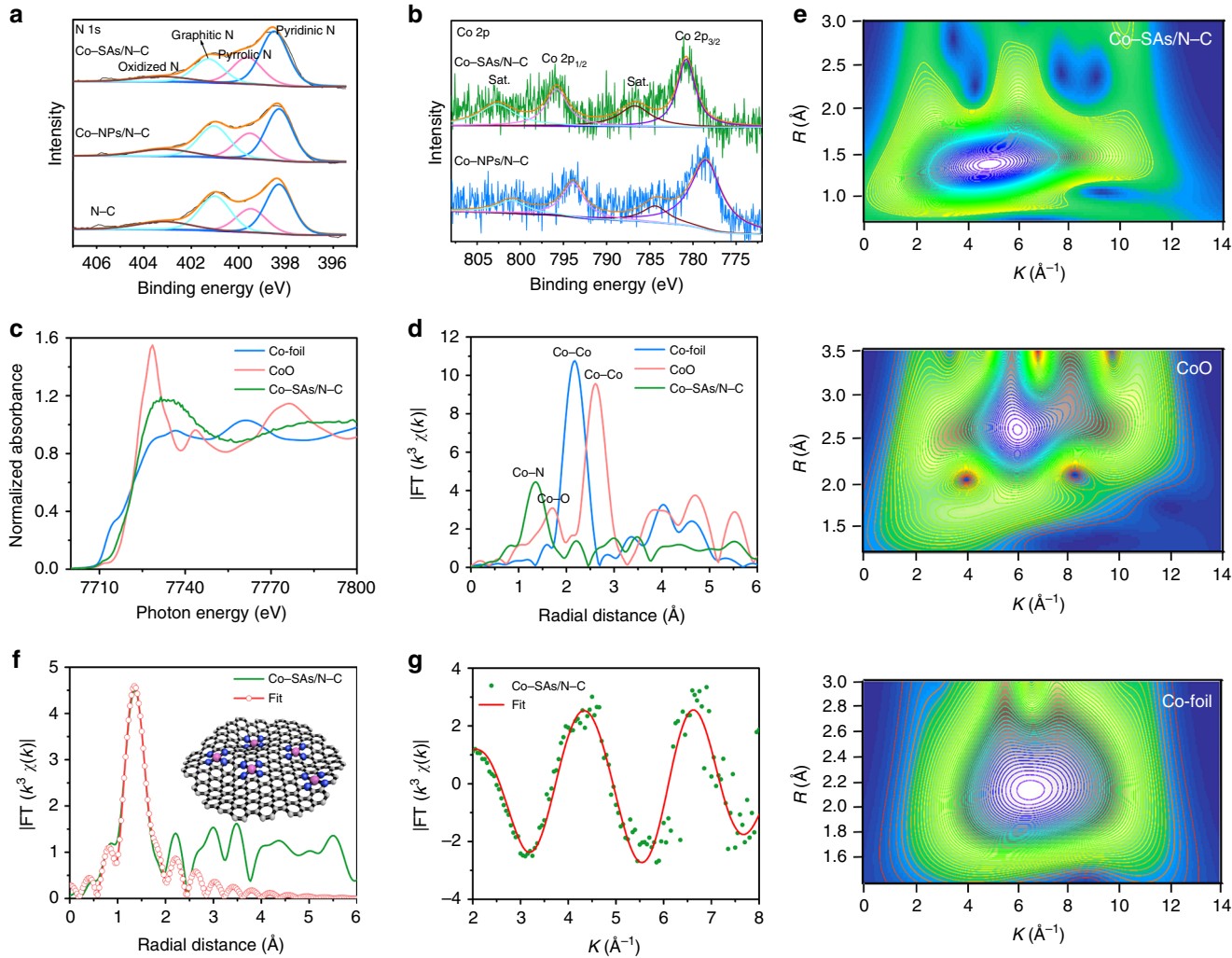

**Fig. 2 XPS and XAFS measurements of catalysts. a** High-resolution XPS N 1s spectra of Co-SAs/N-C, Co-NPs/N-C and N/C. **b** High-resolution XPS Co 2p spectra of Co-SAs/N-C, Co-NPs/N-C and N/C. **c** The normalized K-edge XANES and **d** K-edge FT-EXAFS in R space for Co-SAs/N-C and Co-foil, and CoO reference samples. **e** Wavelet transforms for the k³-weighted EXAFS signals. **f**, **g** Corresponding EXAFS fitting curves at R and K space, respectively, inset showing the schematic model (The pink, bule, and gray balls stand for Co, N, C, respectively).

irreversible capacity loss during the OER process. As revealed in the first cycle's discharge/charge curves (Fig. 3b), it is clearly indicated that the voltage gap characteristic of Co-SAs/N-C electrode delivers much decreased discharge (0.12 V) and charge (0.28 V) polarizations (corresponding to overall potential gap of 0.4 V and ORR/OER efficiency of 87.7%) at a current density of 200 mA g⁻¹ under a curtailing capacity of 1000 mAh g⁻¹. The corresponding voltage gaps for Co-NPs/N-C and N-C cathodes are calculated as 1.02 and 1.03 V, respectively. The outstanding alleviated polarization output of Co-SAs/N-C is comparable to those reported precious metal-based catalysts (Supplementary Table 3). Figure 3c depicts galvanostatic voltage profiles of the three samples during deep discharge/charge process. Apart from elevated discharge plateau, Co-SAs/N-C based Li-O₂ cell also delivers an apparently optimized ORR/OER overpotential of 0.97 V, much smaller relative to Co-NPs/N-C (1.35 V) and N-C (1.28 V) electrodes. Of noteworthy is that, for Co-SAs/N-C, the charge potential gradually reaches a very flat plateau around 3.74 V and conducts in a considerably long capacity interval. In contrast, the charge potential quickly ascends to cutoff voltage (4.5 V) without visible plateau feature for another two samples, indicative of thoroughly different oxygen evolution mechanism. Moreover, Co-SAs/N-C based Li-O₂ cell yields large initial discharge

capacity of 20,105 mAh g⁻¹ and recharge capacity of 19,765 mAh g⁻¹. The corresponding coulombic efficiency closes to 100%, far better than those of Co-NPs/N-C (17,160 mAh g⁻¹, 10,417 mAh g⁻¹, 60.7%) and N-C (17,400 mAh g⁻¹, 10,893 mAh g⁻¹, 62.6%) electrodes. We also investigate the rate capability at various current densities, as depicted in Supplementary Fig. 15a–c, lithium-oxygen cell assembled by Co-SAs/N-C can deliver larger ORR/OER capacity (11,098/7815 mAh g⁻¹) capacity and satisfactory overpotential even at 1 A g⁻¹. The corresponding capacity retentions and coulombic efficiency at 200, 400, and 1000 mA g⁻¹ of the three species are illustrated in Supplementary Fig. 15d, demonstrating the overwhelming advantage in boosting ORR/OER catalytic activity at high rate for Co-SAs/N-C. Meanwhile, we investigate the rate behavior under fixed capacity route. As demonstrated in Fig. 3d and Supplementary Fig. 16, Li-O₂ battery based on Co-SAs/N-C electrode displays both much lower discharge and charge voltage fluctuations. For example, at the case of 200 mA g⁻¹, the terminal ORR and OER voltages are 2.83 and 3.59 V, and when the density value is changed to 1.0 A g⁻¹, the terminal discharge potential slightly decreases to 2.74 V and the charge potential gradually increases to 3.92 V. Nevertheless, Co-NPs/N-C and N-C involved Li-O₂ cells both display more remarkable potential instability with the

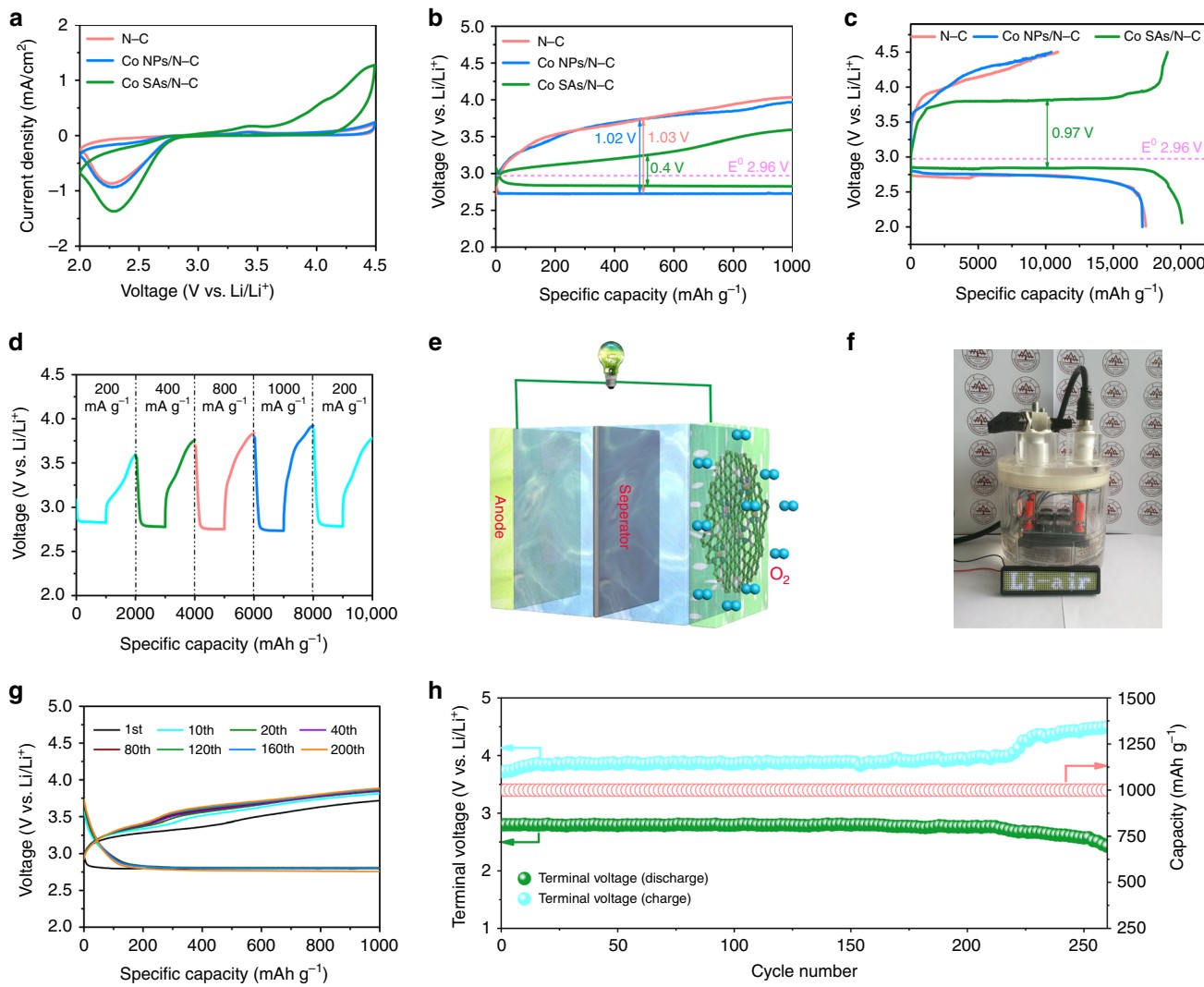

**Fig. 3 Electrochemical performance of Co-SAs/N-C, Co-NPs/N-C and N/C electrodes. a** CV curves of Co-SAs/N-C, Co-NPs/N-C and N/C electrodes at a scan rate of 0.1 mV s⁻¹ at a voltage window of 2.0–4.5 V. **b** The discharge–charge curves of Co-SAs/N-C, Co-NPs/N-C, N-C electrodes at a curtailed capacity of 1000 mAh g⁻¹ at a current density of 200 mA g⁻¹. **c** The initial deep discharge–charge curves of Co-SAs/N-C, Co-NPs/N-C, N-C electrodes at a current density of 200 mA g⁻¹. **d** The discharge–charge profiles of Co-SAs/N-C electrode at various current densities ranging from 0.2 to 1.0 A g⁻¹ at a cut-off capacity of 1000 mAh g⁻¹. **e** The configuration model of a Li-O₂ battery based on Co-SAs/N-C. **f** Photo image of the LED turned on by four as-fabricated Co-SAs/N-C based Li-O₂ batteries. **g** Discharge–charge profiles of Co-SAs/N-C based electrode with different cycles at 400 mA g⁻¹. **h** Cycling stability and terminal discharge–charge voltages of Co-SAs/N-C electrode at 400 mA g⁻¹ with a limited capacity of 1000 mAh g⁻¹.

corresponding discharge/charge voltage reaching about 2.52/4.36 V at 1.0 A g⁻¹. On the other hand, when we set back the current density to primitive value, the ORR/OER potentials exhibit a highly reversible recovery for Co-SAs/N-C, further manifesting its preferable rate reliability. Figure 3e shows a typical configuration model of Co-SAs/N-C-based Li-O₂ battery. Figure 3f exhibits the operation installation of four Li-O₂ button cells based on Co-SAs/N-C electrode in successfully powering a light-emitting diode (LED) array. Long-term cycling performance as another crucial indicator is assessed. The discharge/charge profiles of the Co-SAs/N-C coupled cell for various cycles under 400 mA g⁻¹ (Fig. 3g) exhibit striking overlap ratio, indicating obviously improved cycle stability. Even after 200 cycles, the terminal ORR and OER potentials of Co-SAs/N-C can still maintain at 2.7 and 3.9 V, yielding overall overpotential still at lower level, which is of vital importance in suppressing the intractable parasitic reactions stemmed from electrolyte instability. Figure 3h shows that the 2032 cell based on Co-SAs/N-C can well operate for up to 260 cycles at 400 mA g⁻¹ under a curtailed capacity of 1000 mAh g⁻¹.

Both the lithiation and delithiation potentials can sustain steady over a long period of cycling lifespan, featuring a prominent cycling reliability. In sharp contrast, as seen in Supplementary Fig. 17, 18, Co-NPs/N-C and N-C involved Li-O₂ cells both suffer severe overpotential loss and can only maintain 80 and 60 cycles, respectively, with the OER terminal potential quickly increasing to 4.5 V. Furthermore, we carried out cycling measurements under 400 mA g⁻¹ with extended cut-off capacity of 12,000 mAh g⁻¹ (the corresponding depth of discharge is close to 75%) via the achievable Co-SAs/N-C catalyst, For comparison, the ORR/OER capacities are limited within 3500 mAh g⁻¹ (the discharge depth is close to 35%) for N-C catalysts. As observed in Supplementary Fig. 19a, Co-SAs/N-C based cells can operate very stable for at least 18 cycles under deep charge/discharge processes. In contrast, N-C based cells can only maintain 12 cycles even under much lower cut-off capacity (Supplementary Fig. 19b). This finding further demonstrates the advantage of the Co-SAs/N-C hybrid in dramatically ameliorating overpotential and enhancing cycling durability of lithium-oxygen batteries. Thus, we conclude the

well-distributed $Co-N_4$ configurations play a decisive role in alleviating charge/discharge polarization, especially OER overpotential, leading to greatly improved round-trip efficiency, coulombic efficiency and cycling performance. As seen in Supplementary Table 3, the outstanding electrochemical performance of Co-SAs/N-C places it in advance of some published functional carbonaceous and noble metal catalysts[12,13,53–62].

**Microstructure evolution of electrodes after ORR/OER.** To achieve in-depth unveiling of the ORR and OER working principle associated with $Co-N_4$ active sites, it is of great importance to learn the microstructure evolution of the discharged/recharged cathodes. Ex situ SEM images of the Co-SAs/N-C, N-C and Co-NPs/N-C electrodes after the first full discharge are depicted in Supplementary Figs. 20–22. As depicted in Supplementary Fig. 20, as for Co-SAs/N-C, the carbon skeleton matrices are uniformly covered with numerous discharge products with a diameter of below 10 nm. In contrast, as for Co-NPs/N-C and N-C discharged cathodes displayed in Supplementary Fig. 20b, c, $Li_2O_2$ aggregates with much larger size (c.a. 100–300 nm) have been generated on the N-C platform with an uneven distribution status. Ex situ TEM (Supplementary Figs. 23, 24a, 25a) images of the three species electrodes after the first full discharge are illustrated. For Co-SAs/N-C, uniformly distributed $Li_2O_2$ assemblies are immobilized on the nanosheet surface, allowing for a better interface contact between $Li_2O_2$ and $Co-N_x$ active sites. In sharp contrast, for Co-NPs/N-C and N-C, $Li_2O_2$ aggregates with much larger sizes (above 200 nm) discretely and sparsely deposit on the carbon matrices, which are not easily decomposed during the following recharge process. As depicted in Fig. 4a, detailed microstructure for the $Li_2O_2$ discharged products are revealed, showing uniform fine nano-sized particles about 2−3 nm. The conspicuous lattice fringe can be assigned to (100) plane of $Li_2O_2$ for the discharged Co-SAs/N-C electrode. Furthermore, ex situ SAED (Fig. 4b, Supplementary Figs. 24b, 25b) images of the three discharged electrodes present striking differences for the discharged $Li_2O_2$ products. For Co-SAs/N-C after discharge, diffraction rings associated with (100), (110), and (112) planes of $Li_2O_2$ are much stronger compared to N-C and Co-NPs/N-C. Meanwhile, ex situ XRD results of the three electrodes at different stages are demonstrated in Fig. 4e and Supplementary Fig. 26. Clearly, all the three species depict a distinct peak at about 32.8° well matching with (100) plane of $Li_2O_2$ (JCPDS Card No. 74-0115) after discharge. The peak can reversibly disappear after recharge, indicating the major capacity contribution originated from the generation and oxidation of solid lithium peroxide, consistent with Raman spectra in Supplementary Fig. 27. Surprisingly, the XRD peak of $Li_2O_2$ for Co-SAs/N-C presents two conspicuous differences in detail. One is the obvious peak shift to the low angle position[56,63,64]. The other one is the remarkably enhanced (100) peak intensity, accounting for much better crystallinity of $Li_2O_2$ and larger discharge capacity for Co-SAs/N-C, in line with ex situ SAED results in Fig. 4b. Thus, Co-SAs/N-C catalyst possessing abundant $Co-N_x$ configurations exerts strong influence on tailoring the size, and distribution characteristic of $Li_2O_2$ simultaneously during ORR. In turn, for the optimized catalyst with isolated cobalt, the achieved 2–3 nm nano-sized $Li_2O_2$ with homogeneous contact interface can make full use of the OER catalytic activity of accessible $Co-N_x$ moieties and thus can be more easily to be decomposed, leading to a largely reduced overpotential during recharge. Nevertheless, for N-C and Co-NPs/N-C catalysts, the discrete and random distribution together with the undesired large aggregation characteristic of $Li_2O_2$

species deteriorates its decomposition convenience during recharge. The ex situ TEM (Supplementary Figs. 28–30) images of the three electrodes after the first full recharge directly confirm this point, for Co-SAs/N-C, the discharged products obviously disappear. Crucially, Fig. 4c, d shows Ex situ HAADF-STEM and EDS images of Co-SAs/N-C after the first cycle. It is suggested that the isolated Co atoms can still be maintained without aggregation, exhibiting excellent stability, which guarantees the distinguished redox activity during the whole OER/ORR processes. In sharp contrast, many residues still deposit on the surface of carbon nanosheets of N-C and Co-NPs/N-C electrodes, indicating poor reversibility.

We have conducted differential electrochemical mass spectrometry (DEMS) procedures to monitor the evolved gases during ORR/OER process for the Co-SAs/N-C and N-C based $Li-O_2$ batteries (Supplementary Fig. 31). The DEMS curves confirm the redox processes have been overwhelmingly governed by oxygen expending and releasement, demonstrating considerable reversibility for both the two catalysts. However, relative to N-C, at the end of charging process, less trace amount of $CO_2$ has been generated for Co-SAs/N-C, implying the much enhanced capability of reducing polarization and thus restraining parasitic electrochemistry for the latter[65]. Fourier transform infrared (FTIR) characterizations in Fig. 4f provide powerful evidence that the presence of of $Li_2CO_3$ (860, 1437, 1510 cm$^{-1}$), $CH_3COOLi$ (1197, 1615 cm$^{-1}$) and $HCOOLi$ (1360 cm$^{-1}$) in both Co-NPs/N-C and N-C discharged and charged electrodes during the 10th cycle[14,53,66,67]. Encouragingly, for Co-SAs/N-C, much less side product accumulations are generated over cycles, further indicating its remarkable advantage in boosting ORR/OER reversibility, reducing voltage gap and thus restraining parasitic reactions derived from electrolyte decomposition or erosion with carbonaceous catalyst at high charge overpotential[53]. Careful ex situ XPS proofs during the 10th cycled electrodes have been given as Fig. 4g, h and Supplementary Fig. 32. For Co-SAs/N-C cathode, both Li 1s and C 1s XPS peak characteristics clearly confirm the reversible formation and decomposition of $Li_2O_2$ governs the redox electrochemistry process with negligibly observable signal assigned to parasitic products. Unfortunately, for N-C and Co-NPs/N-C catalytic electrodes, after recharge, the typical byproduct $Li_2CO_3$ signal situated in 55.4 eV dominates in the spectra of Li 1s, and a series of undesirable side products embracing $Li_2CO_3$ and lithium alkyl carbonates are identified in the C 1s spectra[56], in accordance with the above FTIR evidences. Electrochemical impedance spectroscopy (EIS) spectra for the 10th cycled samples (Supplementary Fig. 33) also confirm this point. The impedance of Co-SAs/N-C cathode can well recover to its initial value even after the 10th recharge procedure. However, it is not the case for recharged Co-NPs/N-C and N-C cathodes, the impedance values are still considerably larger than those of fresh electrodes owing to the passivation of active sites arisen from gradually accumulated insulated byproduct residues as cycling continuing. In addition, XRD (Supplementary Fig. 34), HAADF-STEM (Supplementary Fig. 35) and SAED (Supplementary Fig. 36) demonstrate no aggregation of the isolated cobalt atoms even after the 200th cycle within our Co-SAs/N-C catalyst. Furthermore, XPS characterization has been carried out for the 200th cycled Co-SAs/N-C (Supplementary Fig. 37), as compared with the fresh one, the Co 2p XPS characteristics for the Co-SAs/N-C hybrid after electro-catalytic test are largely maintained without attenuation of activity, manifesting stable durability during long-time electrolysis. Herein, we infer that the strengthened long-life robustness of Co-SAs/N-C is assigned to the preferable anchoring interaction of neighboring doped N to isolated Co atoms, effectively suppressing Co aggregation and electrocatalysis deactivation even under the harsh electrolyte conditions.

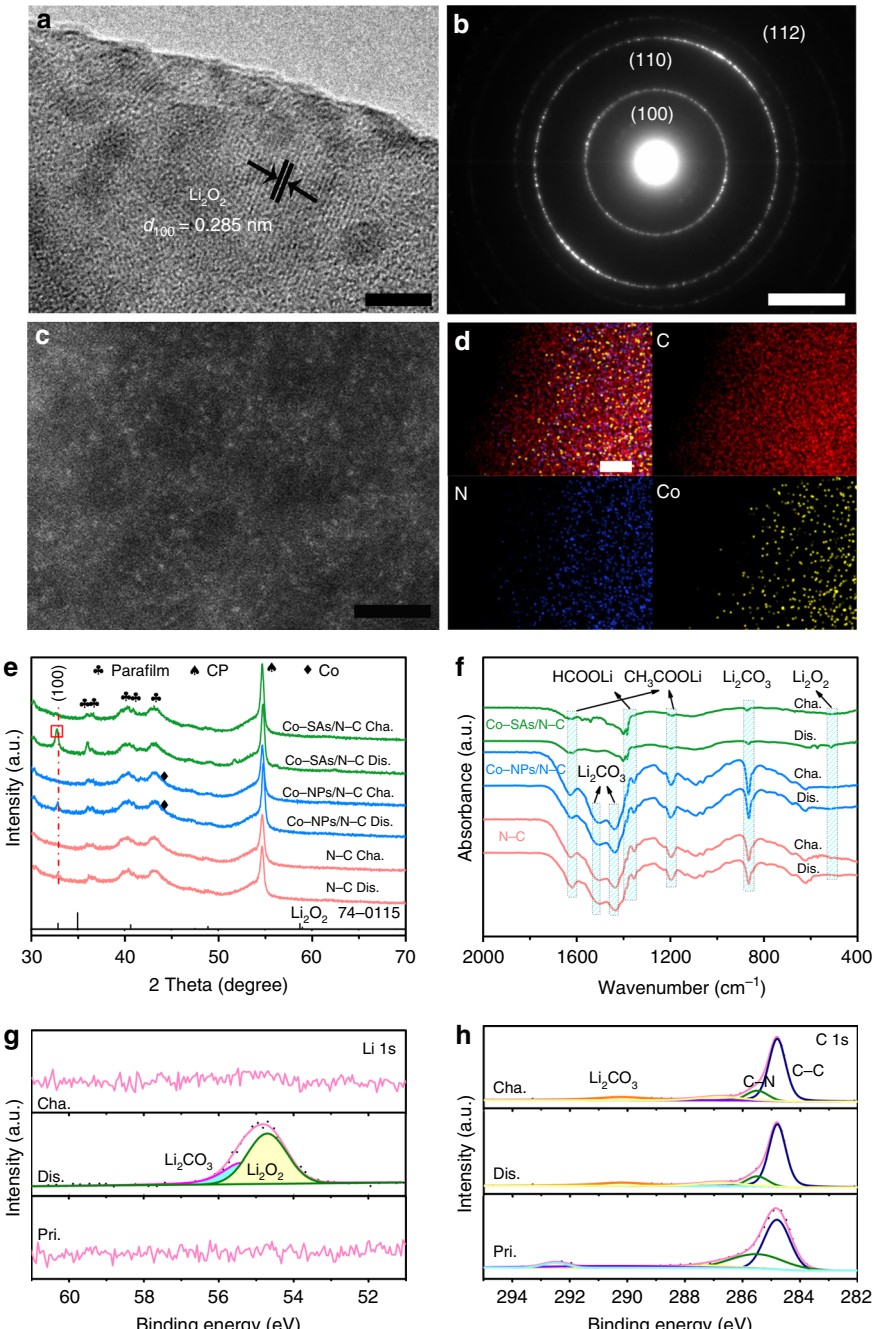

**Fig. 4 Ex situ characterizations of discharged and charged electrodes. a** Ex situ HRTEM image of fully discharged Co-SAs/N-C. **b** Ex situ electron diffraction pattern of fully discharged Co-SAs/N-C. **c**, **d** Ex situ HAADF-STEM and mapping images of Co-SAs/N-C after the 1st cycle. **e** Ex situ XRD patterns of discharged and charged Co-SAs/N-C, Co-NPs/N-C and N/C electrodes during the 1st cycle. **f** Ex situ FTIR spectra of the discharged/charged electrodes at the 10th cycle. **g**, **h** Ex situ XPS spectra of pristine, discharged and recharged Co-SAs/N-C electrode in Li 1s and C 1s regions at the 10th cycle. Pristine, Discharged and Charged are abbreviated as Pri., Dis. and Cha., respectively.

**DFT calculations and proposed mechanism**. To further shed light on the in-depth redox mechanisms and lift the veil of why the electrochemical performance can be substantially boosted by Co-SAs/N-C catalyst, we carry out DFT simulations to learn the corresponding adsorption energy of intermediate products ($LiO_2$) and free energy diagrams during ORR and OER. Figure 5a shows the energetically optimized adsorption construction between $LiO_2$ and Co-SAs/N-C. The Li atom of $LiO_2$ is effectively coordinated with N atom, one O atom of $LiO_2$ is strongly bound with neighboring single Co atomic site. Li and O atoms are simultaneously involved with bonding with Co-$N_4$ configuration,

displaying an intrinsic adsorption energy ($\Delta E_{ads}$) of $LiO_2$ for Co-$N_4$ moiety as high as −8.97 eV. In sharp contrast, as depicted in Fig. 5b, without the Co-$N_4$ configuration, only Li atom of $LiO_2$ is captured by exposed N atoms, leading to a much lower $\Delta E_{ads}$ value of −0.82 eV. The absorption structures of $LiO_2$ on (111), (200), and (220) planes of Co nanoparticle are demonstrated in Fig. 5c and Supplementary Fig. 38, delivering $\Delta E_{ads}$ values of −2.75, −5.09 and −3.31 eV, respectively. The strong binding interaction between Co-SAs/N-C and intermediates critically makes a strong influence on the growth behavior of $Li_2O_2$, which will be discussed later.

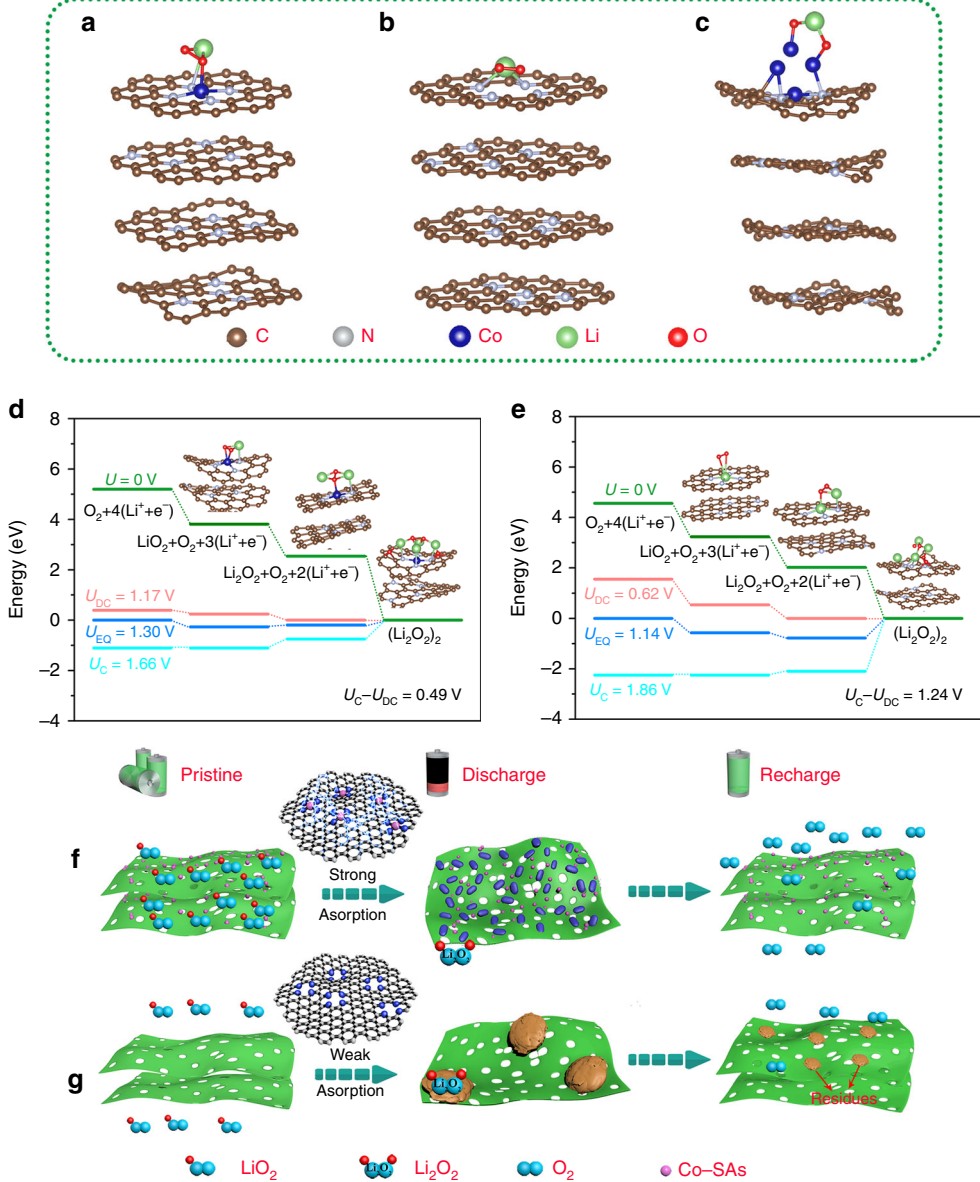

**Fig. 5 DFT calculations and proposed mechanisms. a–c** The optimized structures of $LiO_2$ adsorbed on $Co-N_4$ configuration of Co-SAs/N-C, $LiO_2$ adsorbed on N-C and $LiO_2$ adsorbed on (111) plane of Co nanoparticle in Co-NPs/N-C. **d, e** Calculated free energy diagrams for the discharge–charge reactions on the active surface of Co-SAs/N-C and N-C. **f, g** Schematic illustrations of the working mechanism for the Co-SAs/N-C and N-C electrodes.

Figure 5d, e and Supplementary Fig. 39 give the calculated free energy pathways of discharge and charge processes based on nucleation/decomposition of $(Li_2O_2)_2$ clusters at different overpotentials for the three catalysts. At zero potential, the energy difference between each pathway accounts for the adsorption energies of the various intermediates. The three kinds of catalysts demonstrate similar rate-limiting electrochemical step during discharge and charge, i.e., the limitation steps for the ORR/OER pathways involve the $Li_2O_2$ cluster growth and $LiO_2$ oxidation, respectively. Then, the voltage gaps are investigated to analyze the catalytic reactivity. Here discharge and charge overpotentials are defined as $\Delta_{ORR} = U_{EQ} - U_{DC}$ and $\Delta_{OER} = U_C - U_{EQ}$, respectively, where $U_{EQ}$, $U_{DC}$, and $U_C$ stand for the equilibrium potential, discharge and charge voltage, respectively[13,57]. When coming to Co-SAs/N-C (Fig. 5d), corresponding calculated overpotentials of oxidation reduction and oxidation evolution are 0.13 and 0.36 V, respectively. Nevertheless, the corresponding $\Delta_{ORR}/\Delta_{OER}$ values

for N-C (Fig. 5e) and Co-NPs/N-C (Supplementary Fig. 39) are 0.52/0.72 V and 0.34/0.51 V, respectively, which are in accordance with the experimental results in Fig. 3. Herein, the theoretical calculations support a fact that the exposed $Co-N_4$ active sites embody a decisive factor for reducing both the $Li_2O_2$ generation and oxidization overpotentials.

Integrating computational calculation with experimental data, we clarify a plausible route correlated with $Li_2O_2$ nucleation and growth process. The corresponding schematic illustrations are demonstrated in Fig. 5f, g. Typically, upon ORR, dissolved oxygen initially undergoes a one-electron reduction process, forming intermediate $LiO_2$ ($O_2 + e^- + Li^+ \rightarrow LiO_2$). Afterward, due to the considerably strong constraint effect from the higher affinity between $LiO_2$ and $Co-N_4$ configurations, immobilized $LiO_2^*$ cannot freely assemble, but inevitably wait for extra charge to suffer a second one-electron transfer electrochemical process or chemical disproportionation procedure ($LiO_2^* + e^- + Li^+ \rightarrow Li_2O_2$ or $2LiO_2^* \rightarrow Li_2O_2 + O_2$). As encouraged by plentiful strong adsorption centers and

nucleation sites, homogeneously distributed $Li_2O_2$ accommodations can be formed by this "surface-adsorption pathway". Thus, densely packed $Li_2O_2$ assemblies with nanoscale size can be homogeneously and intimately deposited around active sites, establishing a superior low-impedance $Li_2O_2$/cathode contact interface with good compatibility. During recharge, the unique $Li_2O_2$ taking advantage in ease of decomposition can also make full use of the intimately contacted catalytic centers of Co-$N_4$, thus guaranteeing ultra-low OER overpotential and excellent recovery capability. In sharp contrast, for N-C catalyst, owing to the much weaker binding interactions between $LiO_2$ and limited accessible deposition sites, soluble $LiO_2$ molecules ($LiO_2$ (sol)) can freely migrate and undergo chemical disproportionation route ($2LiO_2(sol) \rightarrow Li_2O_2 + O_2$), thus growing larger and generating isolated $Li_2O_2$ agglomerates, randomly and discretely distributed on the carbon substrate[55]. Trapped in the discrete deposition sites and unsatisfactory OER catalytic characteristic of carbon support, the solid agglomerates cannot be effectively oxidized, resulting in discharge product residues and inferior electrochemical performance. As for Co-NPs/N-C, totally different from the ionic characteristic of Co-$N_4$ configuration, the metallic Co nanoparticles can contribute negligible redox catalytic activity during discharge and charge, further emphasizing the critical role of our single atom catalysts in conjunction with ultrathin carbon nanosheets in maximally exposing active sites and enhancing redox kinetics.

## Discussion

In summary, we first fabricate 2D nitrogen-rich carbon nanosheets coordinated with isolated Co-$N_x$ active sites by a green gas-migration-trapping procedure, acting as a catalyst for lithium-oxygen batteries. Most importantly, the underlying influence mechanisms on tuning geometric structure, size and distribution characteristic of $Li_2O_2$ by single atom catalyst are first systematically investigated. A series of critical characterizations embracing XPS, XAFS and HAADF-STEM confirm the atomic isolation feature of Co-$N_4$. Critically, according to DFT calculations, atomically dispersed Co atoms bonded to N atoms can function as active catalytic centers, drastically enhancing the intrinsic $LiO_2$-absorption ability and thus fundamentally modulating the growth process and distribution of involved $Li_2O_2$. Totally different from bare carbon nanosheets and cobalt nanoparticles counterparts, uniformly distributed nano-sized $Li_2O_2$ species are generated for the Co-SAs/N-C electrode during ORR process. Simultaneously, such uniformly distributed $Li_2O_2$ nanosized assemblies can give full play to the catalytic efficacy of abundant Co-$N_4$ moieties during OER, substantially accelerating $Li_2O_2$ decomposition kinetics and restraining side reactions. As expected, the Co-SAs/N-C electrode can afford ultra-low charge/discharge polarization (0.40 V), superior high rate discharge capacity (11,098 mAh $g^{-1}$ at 1 A $g^{-1}$), excellent cyclability (260 cycles at 400 mA $g^{-1}$). This work opens an avenue to the application of single atom catalysts in the fields of alkali metal-$O_2$ and metal-$CO_2$ batteries.

## Methods

**Material synthesis**. First, 4.0 g hexamine and 8.4 g $Zn(NO_3)_2 \cdot 9H_2O$ were mixed in 200 mL ethyl alcohol solution under vigorous agitation for 0.5 h. Afterward, the transparent solution of $Zn(NO_3)_2 \cdot 9H_2O$ was slowly dropped into the hexamine solution with continuous agitation. The mixed solution was kept still for night under room temperature. Then the white precipitates were achieved after centrifuge and were cleaned by ethyl alcohol and deionized water for three times. Afterward, the obtained samples were vacuum-dried at 80 °C for 12 h, yielding Zn-hexamine white powders[48]. The obtained Zn-hexamine powders (2 g) undergo a simple annealing conducted at 900 °C for 3 h under an argon stream (10 ml min$^{-1}$) with a ramp rate of 5.6 °C min$^{-1}$, yielding N-C black powders (0.28 g). Typically, 0.1 g $CoCl_2 \cdot 6H_2O$ and 0.06 g N-C were laid in two separated zones in a

custom-built porcelain boat (as seen in scheme 1). Then the boat was heat-treated at different temperatures (between 500 and 900 °C) for 1 h under Ar atmosphere under a ramp rate of 5.0 °C min$^{-1}$. After naturally cooling down, the desired catalysts were harvested.

**Material characterization**. The crystallographic characteristics of the species were determined via powder XRD technique employing a diffractometer with Cu $K_\alpha$ radiation (Rigaku D/Max-KA, $\lambda = 1.5406$ Å). The morphology features were obtained from a field emission scanning electron microscope (FE-SEM, JSM-7610 F) and high-resolution transmission electron microscopies (HR-TEM, JEM-2100, 200 kV). Atomic-level high-angle annular dark-field scanning TEM (HAADF-STEM) images were recorded from a probe corrected TEM (FEI, Titan G2 60-300) working at 300 kV, coupled with double probe spherical aberration correctors. The contents of Co element in the samples were tested from inductively coupled plasma optical emission spectroscopy (ICP Optical Emission Spectrometer Varian 720-ES). X-ray photoelectron spectroscopy (XPS, ESCALAB 250 spectra with 150 W Al $K_\alpha$ probe beam) was measured to investigate surface composition and bonding states. X-ray absorption fine structure (XAFS) spectra of the Co K-edge were conducted at room temperature from the beamline 1W1B station (Beijing Synchrotron Radiation Facility, China). A double-crystal Si (111) monochromator was employed to monochromatize the X-ray. Co foil for Co K-edge was used to calibrate the energy. Co foil and CoO were also tested for reference. The XAFS results were fitted via the IFEFFIT software. The specific surface and pore size distribution curves were confirmed using BJ builder Kubo X1000 at 77 K. All the Raman spectra were collected via an in Via-refex micro-Raman spectrometer by a 532 nm laser. FTIR spectroscopy was conducted on a Bruke Vector 22 system.

**Electrochemical measurements**. Prior to assembled, the cathode was fabricated by casting the well agitated slurry, containing 90% catalyst components and 10% polytetrafluoroethylene (PTFE) dissolved in isopropyl alcohol onto carbon paper (13 mm). The catalysts casting was 0.5–0.6 mg cm$^{-2}$. After drying at 80 °C for 24 h, the working cathode dice was achieved. Afterward, the 2032 coin cells were constructed inside an Ar glove box (both $H_2O$ content and $O_2$ content below 1 ppm). Concretely, Whatman glass fiber separator (GF/D) was placed between lithium anode and the achieved catalytic cathode with 180 μL electrolyte (1 M LiTFSI/TEGDME). Prior to measurements, the cells were kept still at open circuit for night in a hermetic container purged with desiccative ultra-pure $O_2$ (99.999%, 1.0 atm). Then, the electrochemical tests were performed in a galvanostatic mode employing a CT2001A LAND battery tester. EIS tests were also conducted at PARSTAT2273 in a frequency range of 0.01–$10^5$ Hz. CV tests were conducted on a PARSTAT2273 electrochemical workstation, scanning between 2.0 and 4.5 V at 0.1 mV s$^{-1}$ speed. All the electrochemical measurements were implemented at 25 °C.

**Computational details**. The Vienna ab initio simulation package (VASP) was utilized to carry out the calculations[68]. The core electrons were treated based on the projector-augmented-wave (PAW) method. The electron interactions were optimized by the Perdew-Burke-Ernzerhof (PBE) generalized gradient approximation[69,70]. The supercell model was established with cell dimensions of $a = 17.88$ Å, $b = 17.92$ Å. The periodic images of the atoms were separated by a vacuum slab of 15 Å in $c$-axis to eliminate the interplay between the periodic images. The convergence criterion for energy was limited at $1 \times 10^{-4}$ eV. The planewave limited energy was controlled at 520 eV. The atomic geometries were relaxed until the threshold forces less than 0.05 eV Å$^{-1}$ during structure optimization. The k-point grid was set as $1 \times 1 \times 1$ for majorization. The adsorption energy ($E_{ads}$) is defined as $\Delta E_{ads} = E_{substrate + LiO_2} - (E_{substrate} + E_{LiO_2})$, where $E_{substrate + LiO_2}$ is the sum energy of the optimized $LiO_2$/substrate system, $E_{substrate}$ and $E_{LiO_2}$ refer to the sum energies of the bare substrate and individual $LiO_2$, respectively. As for Co nanoparticle, we choose three representative crystallographic planes ((111), (200), and (220) planes) of Co nanoparticle with face-centered cubic structure within Co-NPs/N-C catalyst. Considering the interactions between carbon nanosheets and N atoms, Co single atom and Co nanoparticle, the four-layer carbon nanosheets are all relaxed during the structural optimization.

## Data availability

All data employed in this work can be available from the corresponding author upon reasonable request.

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

## Acknowledgements

We acknowledge the projects supported by the State Key Program of National Natural Science of China (No. 51532005, U1906227), the National Nature Science Foundation of China (No. 51802175, 51872171).

## Author contributions

L.Y. dominated the overall investigation. P.W. germinated the initial experimental scheme, performed the experiment and completed the manuscript. Y.R. and Z.Q. initiated the DFT calculations associated with optimized structures of LiO₂ adsorbed on catalysts and calculated free energy diagrams. P.Z. and M.D. helped with the electro-chemical performance measurement, the XAFS and HAADF-STEM analysis. C.L. and D.Z. assisted in schematic illustrations of the working mechanism of the catalysts. Z.Z. contributed to the data analyses and discussions. L.Z. provided TEM, HRTEM, EDX and ex situ TEM, HRTEM characterizations. R.W. helped with the experiment design, result evaluation and manuscript revise. All authors had hand in the modification and improvement of the original manuscript.

## Competing interests

The authors declare no competing interests.
