## [Peer Review File · Nature Communications]

Reviewers' comments:

Reviewer #1 (Remarks to the Author):

This manuscript reported a catalyst Co-SAs/N-C synthesized by a green gas-migration-trapping procedure which can act as bifunctional catalyst for Li-O₂ batteries. However, it has been reported that catalyst is not effective for oxygen evolution reaction but mainly CO₂ evolution (JACS, 2011, 133, 18038-41). To defend their work, the authors should provide evidence if their catalyst is doing O₂ evolution or CO₂ evolution. Otherwise it could be just side reactions.

1. In Figure 3g and 3h, cycling at 400 mA g⁻¹ under a capacity of 1000 mAh g⁻¹ is less than 1/10 of the total capacity (16000 mAh g⁻¹ (Supplementary Figure 14a)). How would the cycle life change when cycling at a more reasonable capacity say 70-80% of the total capacity?
2. More characterization of the discharge product of Li₂O₂ such as Roman is required to confirm the small particle/amorphous Li₂O₂ which is not easy detected by XRD.
3. In Supplementary Figure 14 (a-c), please label the cathode material.

Reviewer #2 (Remarks to the Author):

In this manuscript, Wang et al. prepared an atomic cobalt catalyst supported on N doped carbon, which shows a superior capacity and an excellent stability. The results are interesting, and some points are need to be clared before it is considered to be published.

- (1) The XPS peak fitting is failed in Fig 2a, where the same N species shows a large difference in FWHM, especially pyrrolic N.
- (2) In line 126, page 4, it makes a confusion by the description "the Co K edge XANES in R and k space". And the conclusion is less rigorous with the square-planar Co-N₄ configuration from the Co K edge EXAFS spectra. One of standard references should be put into Table S2 and all of error bars for each parameter should be provided.
- (3) In Fig. 3a, the authors should give more discussion for the CV curves. What's the meaning of the peak around 3.5 V?
- (4) After the discharge, the SEM is more suitable to check the uniform of the Co-SAs/N-C in a large scale.
- (5) How about the electronic state of the Co 2p for Co-SAs/N-C after the long-time cycles, in comparison with the pristine one?
- (6) Some previous works have been published, such as Na-S batteries (Nat. Commun., 2018, 9, 4082), Li-metal batteries (Adv. Energy Mater., 2019, 9, 1804019) and Li-S batteries (J. Am. Chem. Soc., 2019, 141, 3977-3985; Adv. Mater., 2019, 31, 1903955). It should be summarized and discussed in the introduction section.
- (7) The details of DFT calculation is missing. Most important, the description of the model used in current study is not described. Therefore, the validity and reliability of current calculation can not be evaluated. For example, in Figure 30 in supporting information, there is a huge deformation of graphene, probably because authors used a bad model. Meanwhile, the huge adsorption energy of LiO₂ on Co-N is strange, which means the active site of Co-N will be "poisoned" due to the adsorption of LiO₂. Again, it is not professional to describe the adsorption energy with four number after decimal point. Does the calculated accuracy reach the level of meV?

Reply to Reviewers' comments:

Response to Reviewer #1

Comment 1: This manuscript reported a catalyst Co-SAs/N-C synthesized by a green gas-migration-trapping procedure which can act as bifunctional catalyst for Li-O₂ batteries. However, it has been reported that catalyst is not effective for oxygen evolution reaction but mainly CO₂ evolution (JACS, 2011, 133, 18038-41). To defend their work, the authors should provide evidence if their catalyst is doing O₂ evolution or CO₂ evolution. Otherwise it could be just side reactions.

Response 1: We would like to thank the reviewer for the valuable suggestion. Following the reviewer's comment, we have conducted the quantitative measurements of in-situ DEMS to monitor the evolved gas species during discharge and charge processes, which can offer direct evidence about the reversibility. The DEMS curves are shown below in **Supplementary Fig. 30**. The in-situ DEMS results confirm that the discharge process is overwhelmingly dominated by oxygen rather than CO₂ release. Together with the XRD (**Fig. 4e**), Raman (**Supplementary Fig. 26**), XPS (**Fig. 4g, h**) results, we can confirm that reactions at the cathode during cycling are dominated by reversible Li₂O₂ formation and decomposition with negligible electrolyte decomposition and parasitic reactions for Co-SAs/N-C catalyst.

We have carefully read the literature mentioned above (JACS, 2011, 133, 18038-41), and fully aware that a preferable matching between aprotic electrolyte and Li-O₂ system plays a key role in maintaining electrolyte stability and minimizing parasitic reaction during long-period cycling. In this article, A. C. Luntz et al employed 1PC:1DME and DME as electrolytes separately to investigate gas evolution from cells by DEMS method. The results showed that in spite of Li₂O₂ dominated as discharge products, unsatisfactory CO₂ evolution inevitably conducted during charge. S. A. Freunberger and P. G. Bruce et al also identified the poor rechargeability of the organic carbonates (such as PC) electrolyte.^[R1] L. F. Nazar et al reported the serious decomposition behavior of dimethoxyethane (DME).^[R2] Herein, the stability of PC and DME based electrolytes can not be sustained on cycling, stemming from the nucleophilic attack by the reduced O₂ species with serious byproducts accumulation at the cathode, as shown in the above-mentioned articles. Recently, more reliable tetra (ethylene) glycol dimethyl ether (TEGDME) was developed with low sensitivity towards reactive oxygen radical anion and had been widely used in aprotic Li-O₂ battery. Moreover, several reported works have also offered the evidence of the good stability of the TEGDME electrolyte upon cycling and the reversibility of Li₂O₂ formation-dissolution during discharge/charge processes. Many DEMS characterizations demonstrated the negligible CO₂ evolution during the recharge processes.^[R3-R7] So our DEMS results based on TEGDME system could offer another solid evidence about the relative stability of this electrolyte.

Moreover, given the instability of the electrolyte at high potential, it should be emphasized that controlling charge voltage under 4.0 V is another critical strategy in efficiently maintaining the stability of TEGDME and restraining parasitic electrochemistry especially during long-period cycling. Commonly, as shown in our work, electrocatalysts with excellent catalytic activity play a

vital role in reducing charge polarization, which is beneficial for mitigating the unfavorable side actions. Particularly, to make sure the absolute recovery of discharge products and ameliorate the oxygen reduction/evolution reversibility, controlling discharge/charge depth is widely used in Li-O₂ batteries system. This strategy also aims at reducing the risk of electrolyte decomposition, which is discussed in the following Response 2 part.

All in all, based on the in-situ DEMS studies and the above-mentioned analysis, we conclude that the superior capacity outputs in this work mainly originate from the desirable oxygen evolution reaction instead of side products decomposition during recharge process.

[R1] S. A. Freunberger, Y. Chen, Z. Peng, J. M. Griffin, L. J. Hardwick, F. Barde, P. Novak, P. G. Bruce, *J. Am. Chem. Soc.* **2011**, 133, 8040.

[R2] B. D. Adams, R. Black, Z. Williams, R. Fernandes, M. Cuisinier, E. J. Berg, P. Novak, G. K. Murphy, L. F. Nazar, *Adv. Energy Mater.* **2015**, 5, 1400867.

[R3] Z. Huang, H. Zeng, M. Xie, X. Lin, Z. Huang, Y. Shen, Y. Huang, *Angew. Chem. Int. Ed.* **2018**, 131, 2367.

[R4] R. A. Wong, C. Yang, A. Dutta, M. O. M. Hong, M. L. Thomas, K. Yamanka, T. Ohta, K. Waki, H. R. Byon, *ACS Energy Lett.* **2018**, 3, 592.

[R5] K. R. Yoon, K. Shin, J. Park, S. H. Cho, C. Kim, J. W. Jung, J. Y. Cheong, H. R. Byon, H. M. Lee, I. D. Kim, *ACS nano* **2018**, 12, 128.

[R6] Y. Chang, S. Dong, Y. Ju, D. Xiao, X. Zhou, L. Zhang, X. Chen, C. Shang, L. Gu, Z. Peng, G. Cui, *Adv. Sci.* **2015**, 2, 1500092.

[R7] B. Sun, L. Guo, Y. Ju, P. Munroe, E. Wang, Z. Peng, G. Wang, *Nano Energy* **2016**, 28, 486.

Also, in the manuscript, corresponding revisions have been made. The measured DEMS curves during discharge/charge processes of Co-SAs/N-C and N-C based Li-O₂ batteries have been added as Supplementary Fig. 30 on Page 18 in the revised Supplementary Information. The description has been added on Page 7 in the revised manuscript.

Supplementary Figure 30. Differential electrochemical mass spectrometry (DEMS) analysis of the evolved gases during the a) discharge and b) charge of a Li-O₂ cell with Co-SAs/N-C cathode. DEMS analysis of the evolved gases during the c) discharge and d) charge of a Li-O₂ cell with N-C cathode. The solid black lines indicate the potential, while the green and blue solid dots denote the O₂ and CO₂ evolution profiles, respectively.

Line 18 in page 7:

In situ differential electrochemical mass spectrometry (DEMS) analyses have been conducted to monitor the evolved gases during discharge/charge for the Co-SAs/N-C and N-C based Li-O₂ batteries (**Supplementary Fig. 30**). The DEMS results confirm that the redox processes are overwhelmingly dominated by oxygen consumption and release, demonstrating considerable reversibility for both the two catalysts. However, compared with that of N-C, at the end of charging process, less trace amount of CO₂ has been generated for Co-SAs/N-C, implying the much enhanced capability of reducing polarization and thus restraining parasitic electrochemistry for the latter.

Comment 2: In Figure 3g and 3h, cycling at 400 mA g⁻¹ under a capacity of 1000 mAh g⁻¹ is less than 1/10 of the total capacity (16000 mAh g⁻¹ (Supplementary Figure 14a)). How would the cycle life change when cycling at a more reasonable capacity say 70-80% of the total capacity?

Response 2: We would like to thank the reviewer for the constructive comments. The reviewer has provided a very meaningful insight which we will surely pay attention to in the future work. We performed additional cycling measurements with extended capacity for the Co-SAs/N-C catalyst from 1000 mAh g⁻¹ to 12000 mAh g⁻¹ (the corresponding depth of discharge is close to 75%). In contrast, the discharge/charge capacities are limited at 3500 mAh g⁻¹ (considering the much inferior ORR/OER reversibility, the corresponding depth of discharge is close to 35%) for N-C

catalyst, in order to further illustrate the advantage of the Co-SAs/N-C catalyst for dramatically ameliorating overpotential and enhancing cycling stability of Li-O₂ batteries. As depicted in **Supplementary Fig. 18a**, Co-SAs/N-C based cells can operate very stable for at least 10 cycles under deep charge/discharge processes. By contrast, N-C based cells can only maintain 5 cycles even under much lower cut-off capacity (**Supplementary Fig. 18b**).

Commonly, although Li-O₂ batteries reported in most publications can provide the specific capacity as high as several thousand or even over ten thousand mAh g⁻¹, the cycles are very limited when the cells are operated at such high capacity values. The underlying idea means the large rechargeable capacity obtained by deep discharge and charge cannot stand during cycling tests.^[R1-R2] The cathode surface would be thoroughly passivated by insulated discharge products which could not be completely electrochemically oxidized during long cycle operation, triggering higher overpotential and inducing parasitic reactions associated with electrolyte decomposition. In order to address this limitation and extend the cycle operation, reduction of the depth of discharge/charge (the capacities are commonly limited at 500, 1000 or 2000 mAh g⁻¹) to alleviate detrimental side reactions and facilitate the decomposition of solid Li₂O₂ products has been widely used in Li-O₂ batteries.^[R2-R8] In conclusion, a trade-off exists between high capacity and cycling stability for Li-O₂ batteries. The capacities and depth of discharge/charge of some catalytic cathodes reported recently are listed as follows:

Table R1. The capacities and depth of discharge/charge of some catalytic cathodes for Li-O₂ batteries reported recently

Catalysts materials	Fully discharge capacity (mAh g ⁻¹)	Cut-off capacity (mAh g ⁻¹)	Reference
Co ₃ O ₄	9800	1000	R3
MnO ₂ /CNTs	28517	1000	R4
Pt/Pt ₃ Co/Graphene	10000	1000	R5
Se/MnO ₂	~10000	1000	R6
Co[Co,Fe]O ₄ /NG	13312	1000	R7
CNT films	13564	500	R8

[R1] X. Guo, P. Liu, J. Han, Y. Ito, A. Hirata, T. Fujita, M. Chen, *Adv. Mater.* **2015**, 27, 6137.

[R2] X. Guo, N. Zhao, *Adv. Energy Mater.* **2013**, 3, 1413.

[R3] Z. L. Jiang, G. L. Xu, Z. Yu, T. H. Zhou, W. K. Shi, C. S. Luo, H. J. Zhou, L. B. Chen, W. J. Sheng, M. Zhou, L. Cheng, R. S. Assary, S. G. Sun, K. Amine, H. Sun, *Nano Energy* **2019**, 64, 103896.

[R4] L. Ma, N. Meng, Y. Zhang, F. Lian, *Nano Energy* **2019**, 58, 508.

[R5] G. Tan, L. Chong, C. Zhan, J. Wen, L. Ma, Y. Yuan, X. Zeng, F. Guo, J. E. Pearson, T. Li, T. Wu, D. J. Liu, R. S. Yassar, J. Lu, C. Liu, K. Amine, *Adv. Energy Mater.* **2019**, 9, 1900662.

[R6] T. H. Gu, D. A. Agyeman, S. J. Shin, X. Jin, J. M. Lee, H. Kim, Y. M. Kang, S. J. Hwang, *Angew. Chem., Int. Ed.* **2018**, 57, 15984.

[R7] Y. Gong, W. Ding, Z. Li, R. Su, X. Zhang, J. Wang, J. Zhou, Z. Wang, Y. Gao, S. Li, P. Guan, Z. Wei, C. Sun, *ACS Catal.* **2018**, 8, 4082.

[R8] Z. Huang, Z. Deng, Y. Shen, W. Chen, W. Liu, M. Xie, Y. Li, Y. Huang, *J. Mater. Chem. A* **2019**, 7, 3000.

We have added corresponding description in the revised manuscript. The discharge-charge curves

during cycling of Co-SAs/N-C and N-C based Li-O₂ batteries have been added as Figure S18 on Page 12 in the revised Supplementary Information.

Supplementary Figure 18. a) The discharge-charge profiles of Co-SAs/N-C with different cycles with a cut-off capacity of 12000 mAh g⁻¹ at 400 mA g⁻¹, b) N-C based electrodes with different cycles with a cut-off capacity of 3500 mAh g⁻¹ at 400 mA g⁻¹.

Line 4 in page 6:

Furthermore, we carried out cycling measurements with extended cut-off capacity of 12000 mAh g⁻¹ at 400 mA g⁻¹ (the corresponding depth of discharge is close to 75%) for the Co-SAs/N-C catalyst. For comparison, the discharge/charge capacities are limited at 3500 mAh g⁻¹ (the discharge depth is close to 35%) for N-C catalyst. As depicted in **Supplementary Fig. 18a**, Co-SAs/N-C based cells can operate very stable for at least 10 cycles under deep charge/discharge processes. In contrast, N-C based cells can only maintain 5 cycles even under much lower cut-off capacity (**Supplementary Fig. 18b**). This finding further demonstrates the advantage of the Co-SAs/N-C catalyst in dramatically ameliorating overpotential and enhancing cycling stability of Li-O₂ batteries.

Comment 3: More characterization of the discharge product of Li₂O₂ such as Roman is required to confirm the small particle/amorphous Li₂O₂ which is not easy detected by XRD.

Response: Following the reviewer's suggestions, we have carried out Raman spectroscopic measurements of the discharge and charged electrodes based on Co-SAs/N-C and N-C catalysts. The Raman spectra for the discharged/recharged Co-SAs/N-C and N-C electrodes at first full cycle are shown in **Supplementary Fig. 26**. The intense peak at ~798 cm⁻¹ which is characteristic for Li₂O₂, demonstrates that Li₂O₂ dominates the discharge products. After a following recharge, the Li₂O₂ peak vanishes, further confirming an excellent reversibility associated with the desirable formation and decomposition of Li₂O₂, which is in accordance with the XRD results in **Fig. 4e**.

Supplementary Figure 26. Raman spectra of the discharged/charged electrodes for Co-SAs/N-C, and N-C at the 1st full cycle.

We have added corresponding description in the revised manuscript and supporting information.

Line 38 in page 6:

The peak can reversibly disappear after recharge, indicating the major capacity contribution originated from the formation and decomposition of solid Li₂O₂, which is consistent with Raman spectra in **Supplementary Fig. 26**.

Page 14 in the supporting information:

The intense peak at ~798 cm⁻¹ which is characteristic for Li₂O₂, demonstrates that Li₂O₂ dominates the discharge products. After a following recharge, the Li₂O₂ peak vanishes, further confirming an excellent reversibility associated with the desirable formation and decomposition of Li₂O₂, which is in accordance with the XRD results in Figure 4(e).

Comment 4: In Supplementary Figure 14 (a-c), please label the cathode material.

Response: Thank you for the helpful suggestion. We have labeled the three cathode catalysts accordingly in **Supplementary Fig.14 a-c**.

Response to Reviewer #2

In this manuscript, Wang et al. prepared an atomic cobalt catalyst supported on N doped carbon, which shows a superior capacity and an excellent stability. The results are interesting, and some points are need to be clared before it is considered to be published.

Comment 1: The XPS peak fitting is failed in Fig 2a, where the same N species shows a large difference in FWHM, especially pyrrolic N.

Response 1: We would like to thank the reviewer for the valuable suggestion. We have refitted the XPS figures and focused on the FWHM consistency for the same N species. The corresponding revision has been made in Fig 2a in the manuscript (Page 15). The XPS N 1s spectrum is deconvoluted into pyridinic (~398.3 eV), pyrrolic (~399.5 eV), graphitic (~401.3 eV), and oxidized (~403.2 eV)-like N species. It is worthy pointing out that the strongest N 1s peak, attributed to pyridinic N of Co-SAs/N-C, up-shifts by ~0.3 eV with respect to its position in Co-NPs/N-C and N-C.

Fig. 2 a) High-resolution XPS N 1s spectra of Co-SAs/N-C, Co-NPs/N-C and N/C. **b)** High-resolution XPS Co 2p spectra of Co-SAs/N-C, Co-NPs/N-C and N/C. **c)** The normalized K-edge XANES and **d)** K-edge FT-EXAFS in R space for Co-SAs/N-C and Co-foil, and CoO reference samples. **e)** Wavelet transforms for the k^3 -weighted EXAFS signals. **f, g)** Corresponding EXAFS fitting curves at R and K space, respectively, inset showing the schematic model (The pink, blue, and gray balls stand for Co, N, C, respectively).

Comment 2: In line 126, page 4, it makes a confusion by the description “the Co K edge XANES in R and k space”. And the conclusion is less rigorous with the square-planar Co-N₄ configuration

from the Co K edge EXAFS spectra. One of standard references should be put into Table S2 and all of error bars for each parameter should be provided.

Response 2: We would like to thank the reviewer for the valuable suggestion.

i) We have corrected the description about “the Co K-edge XANES in R and k space” as “the Co K-edge EXAFS in R and k space” and rewritten this part for clear understanding.

ii) The quantitative coordination configuration of Co atom can be obtained by EXAFS fitting. The best-fitted result of the EXAFS data is summarized in Table 2 in the supporting information, identifying the first Co-N shell with a coordination number of 3.8 at a distance of 1.94 Å. Moreover, the pre-edge peak at 7711.0 eV for Co-SAs/N-C in Figure 1c is derived from a $1s-4p_z$ transition, demonstrating the fingerprint of Co-N₄ square-planar coordination^{23, 51, 52}. In a work associated with carbon nanosheet loading Co-N₄ species, Y. D. Li et al conducted a DFT calculation on the formation energies of various possible structures to confirm the stability of CoN_x (x =0, 1, 2, 3, and 4) and to testify whether CoN₃, CoN₂, CoN, or CoC exist in the carbon matrix. They concluded that the CoN₄ coordination is the most stable species among different Co-N_x structures.⁵¹ Based on these fitting results, we infer that the local structure of Co-SAs/N-C involves four-fold coordination by N atoms, forming Co-N₄ planar structure.

iii) Furthermore, the coordination structure information of the two standard references (Co foil and CoO) and error bars have been added in Table 2 in the supporting information.

Line 17 in page 4:

As shown in Fig. 2f, the Co K-edge EXAFS fitting of the first shell in R space can be well performed following the Co-N scattering paths. Based on this route, the fitted curve in k space (Fig. 2g) is also well-matched with the experimental data. Then, to achieve the quantitative bonding information, we extract bond lengths and metal coordination numbers from Co K-edge EXAFS curve fitting. As shown in Table 2 in the supporting information, the coordination number (N) of the isolated Co atoms in Co-SAs/N-C is 3.8. Together with the pre-edge characteristic peak in Fig. 1c, we infer that the four-coordinated CoN₄ configurations are formed in the Co-SAs/N-C catalyst, which is consistent with those reported works^{30, 40, 51, 52}.

Two new articles have been cited in the manuscript as Ref. 51-52, respectively.

51. Zhu, Y. et al. A cocoon silk chemistry strategy to ultrathin N-doped carbon nanosheet with metal single-site catalysts. *Nat. Commun.* 9, 3861 (2018).

52. Sun, X. et al. High-performance single atom bifunctional oxygen catalysts derived from ZIF-67 superstructures. *Nano Energy* 61, 245-250 (2019).

Supplementary Table 2. Co K-edge EXAFS curve fitting parameters. ($S_0^2=0.90$)

Catalyst	Shell	N	R(Å)	$\sigma^2 (\times 10^{-3} \text{Å}^2)$	ΔE_0 (eV)	R%
Co-SAs/N-C	Co-N	3.8	1.94	4.0	-2.3	1.0
Co foil	Co-Co	12.0	2.48	6.9	-5.8	0.49
CoO	Co-Co	12.0	3.09	8.6	3.9	0.87
	Co-O	6.0	2.17	7.4	5.3	0.72

N: coordination number;

R: interatomic distance between central atoms and backscatter atoms;

σ^2 : Debye-Waller factor to characterize both thermal and structural disorders;

ΔE_0 : inner potential shift;

R%: the indicator for the goodness of the fit.

S_0^2 : the amplitude reduction factor.

Error bounds (accuracies) are estimated as N, ± 0.1 ; R, ± 0.02 ; σ^2 , ± 1.60 ; ΔE_0 , ± 0.50 .

Comment 3: In Fig. 3a, the authors should give more discussion for the CV curves. What's the meaning of the peak around 3.5 V?

Response: We thank the reviewer for the constructive suggestion. We have revised the discussion for the CV curves in the manuscripts.

Line 35 in page 4:

During the cathodic sweep, the three catalysts show similar reduction peak (2.32 V), corresponding to the occurrence of ORR related to Li_2O_2 product ($\text{O}_2 + 2\text{Li}^+ + 2\text{e}^- \rightarrow \text{Li}_2\text{O}_2$). Meanwhile, during the anodic scan, a clear oxidation peak located at 3.3 V can also be noticeable for the three catalysts, which is associated with the bulk decomposition process for the discharge species ($\text{Li}_2\text{O}_2 \rightarrow \text{O}_2 + 2\text{Li}^+ + 2\text{e}^-$).

However, compared with that of Co-NPs/N-C and N-C, much higher anodic peak current and large integral area can be achieved for the Co-SAs/N-C electrode, implying that more discharge products can be reversibly decomposed and thus much ameliorated catalytic kinetics performance.

Comment 4: After the discharge, the SEM is more suitable to check the uniform of the Co-SAs/N-C in a large scale.

Response 4: Following the reviewer's suggestion, we have performed the post-mortem SEM characterization on the discharged Co-SAs/N-C, Co-NPs/N-C and N-C electrodes during the first cycle, and the SEM images are shown below in Supplementary Fig. 19-21 in the supporting information. As depicted in Figure Supplementary Figure 19, as for Co-SAs/N-C, the carbon skeleton matrices are uniformly covered with numerous discharge products with nano size, establishing a superior contact interface between Li_2O_2 species and N-C nanosheets. In contrast, as for Co-NPs/N-C and N-C discharged electrodes shown in Supplementary Fig. 20 and 21, Li_2O_2 aggregates with larger size (c.a. 100-300 nm) are formed on the N-C substrates surface with an irregularly distribution state, resulting in a much lower coverage rate. The randomly dispersed Li_2O_2 species result in inferior interface between discharge products and active sites, leading to a significant overpotential gap and low catalytic efficiency when charged in turn.

Supplementary Figure 19. EX-situ SEM images of discharged Co-SAs/N-C electrode during the first cycle.

Supplementary Figure 20. EX-situ SEM images of discharged Co-NPs/N-C electrode during the first cycle.

Supplementary Figure 21. EX-situ SEM images of discharged N-C electrode during the first cycle.

We have added corresponding description in the revised manuscript.

Line 19 in page 6:

Ex situ SEM images of the Co-SAs/N-C, N-C and Co-NPs/N-C electrodes after the first full discharge are depicted in **Supplementary Fig. 19-21**. As depicted in **Supplementary Fig. 19**, as for Co-SAs/N-C, the carbon skeleton matrices are uniformly covered with numerous discharge products with a diameter of below 10 nm. In contrast, as for Co-NPs/N-C and N-C discharged electrodes shown in **Supplementary Fig. 20, 21**, Li_2O_2 aggregates with much larger size (c.a. 100-300 nm) are formed on the N-C surface with an irregularly distribution state.

Comment 5: How about the electronic state of the Co 2p for Co-SAs/N-C after the long-time cycles, in comparison with the pristine one?

Response 5: Following the reviewer's comment, to further prove the stability of catalyst, the electronic state of Co-SAs/N-C catalyst after the 200th cycle is also characterized by XPS, and the XPS spectra are shown below as Supplementary Fig. 36 in the supporting information. As compared with the fresh one, the Co 2p XPS spectra for the Co-SAs/N-C catalyst after the electro-catalytic test are largely maintained. Both the peak intensities and positions of Co 2p_{1/2} (~795.8 eV) and Co 2p_{3/2} (~780.8 eV) demonstrate no discernible change before and after 200 continuous ORR/OER cycles, elucidating that the isolated Co atoms can still maintain positively charged state. Accordingly, the post-electrolysis Co 2p XPS spectrum suggests the chemical state of Co-N bond remained without disturbance, manifesting stable durability during long-time electrolysis. Summing up, together with the XRD (Supplementary Fig. 33), HAADF-STEM (Supplementary Fig. 34) and SAED (Supplementary Fig. 35) images, we infer that the high stability of Co-SAs/N-C is attributed to the excellent anchoring effect of neighboring doped N to

Co atoms, effectively suppressing Co aggregation and electrocatalysis deactivation, which also indicates the direct correlation of the superior cycling performance with our single atom catalyst.

Supplementary Figure 36. XPS patterns of the pristine Co-SAs/N-C and the Co-SAs/N-C catalyst scratched from the electrode after the 200th cycle.

We have added corresponding description in the revised manuscript.

Line 4 in page 8:

Furthermore, the Co-SAs/N-C catalyst after the 200th cycle is also characterized by XPS (Supplementary Fig. 36), as compared with the fresh one, the Co 2p XPS spectra for the Co-SAs/N-C catalyst after the electro-catalytic test are largely maintained without attenuation of activity, manifesting stable durability during long-time electrolysis. Herein, we infer that the strengthened long-life robustness of Co-SAs/N-C is assigned to the preferable anchoring interaction of neighboring doped N to isolated Co atoms, effectively suppressing Co aggregation and electrocatalysis deactivation even under the harsh electrolyte conditions.

Comment 6: Some previous works have been published, such as Na-S batteries (Nat. Commun., 2018, 9, 4082), Li-metal batteries (Adv. Energy Mater., 2019, 9, 1804019) and Li-S batteries (J. Am. Chem. Soc., 2019, 141, 3977-3985; Adv. Mater., 2019, 31, 1903955). It should be summarized and discussed in the introduction section.

Response 6: We have carefully read these four articles associated with the pioneering applications of single atoms in secondary battery systems, embracing Li-S, Na-S and Li-metal batteries. We have cited these articles and added the corresponding summary in the introduction section in the revised manuscript:

Line 12 in page 2:

Relative to solid nanoparticles, owing to the super-high utilization of active atoms, non-saturated atomic coordination sites and uniformity of active centers, single-atoms (SAs) species as “Holy Grail” deservedly stand out and occupy research frontier in massive catalytic systems embracing water splitting²¹⁻²⁶, fuel cells^{27,28}, zinc-air batteries²⁹⁻³², nitrogen reduction³³ and CO₂ reduction reactions³⁴⁻³⁹. Especially, in the fields of rechargeable aprotic battery technologies, atomically dispersed metals as active sites also begin to demonstrate considerable potential in accelerating the conversion kinetics and boosting active species utilization efficiently. For example, when introduced to Li and Na-sulphur batteries, single-atoms can conspicuously suppress the “shuttle effect” and electrocatalyze the polysulfide reduction⁴⁰⁻⁴². Moreover, Gong’s group reported that when applied in Li metal anodes, the isolated metal atoms could also affect Li deposition and alleviate dendritic Li growth during the cycles⁴³.

The four articles mentioned above have been cited in the manuscript as Ref. 41-44, respectively.

40. Du, Z. et al. single nickel atoms on nitrogen-doped graphene enabling enhanced kinetics of lithium-sulfur batteries. *J. Am. Chem. Soc.* 141, 3977-3985 (2019).

41. Zhang, L. et al. cobalt in nitrogen-doped graphene as single-atom catalyst for high-sulfur content lithium-sulfur batteries. *Adv. Mater.* 31, 1903955 (2019).

42. Zhang, B. W. et al. atomic cobalt as an efficient electrocatalyst in sulfur cathodes for superior room-temperature sodium-sulfur batteries. *Nat. Commun.* 9, 4082 (2018).

43. Zhai, P. et al. uniform lithium deposition assisted by single-atom doping toward high-performance lithium metal anodes. *Adv. Energy Mater.* 9, 1804019 (2019).

Comment 7: The details of DFT calculation is missing. Most important, the description of the model used in current study is not described. Therefore, the validity and reliability of current calculation can not be evaluated. For example, in Figure 30 in supporting information, there is a huge deformation of graphene, probably because authors used a bad model. Meanwhile, the huge adsorption energy of LiO₂ on Co-N is strange, which means the active site of Co-N will be "poisoned" due to the adsorption of LiO₂. Again, it is not professional to describe the adsorption energy with four number after decimal point. Does the calculated accuracy reach the level of meV?

Response 7: We appreciate the reviewer to bringing up these points.

i) We have revised the details of DFT calculation in the Experimental Section part in the supporting information.

Page 2 in the supporting information:

All the calculations were performed on the basis of DFT implemented in the Vienna Ab initio Simulation Package (VASP).^[S1] The projector-augmented-wave (PAW) pseudopotential was utilized to treat the core electrons, while the Perdew-Burke-Ernzerhof (PBE) exchange correlation functional of the generalized gradient approximation was used to describe electron interactions.^[S2, S3] The supercell model was established with cell dimensions of a= 17.88 Å, b=17.92 Å. The periodic images of the atoms were separated by a vacuum slab of 15 Å in c-direction to avoid the artificial interactions between the periodic images. The cut-off kinetic energy was 520 eV and the

convergence criterion of the total energy was set to be within 1×10^{-4} eV. All atoms were relaxed until the forces were less than 0.02 eV \AA^{-1} during structure optimization. The reciprocal space was sampled using a $1 \times 1 \times 1$ k-point for geometry optimization. The adsorption energy (E_{ads}) is defined as follows: $\Delta E_{\text{ads}} = E_{\text{Substrate+LiO}_2} - (E_{\text{Substrate}} + E_{\text{LiO}_2})$, where $E_{\text{Substrate+LiO}_2}$ is the total energy of the optimized substrate/LiO₂ adsorption system, $E_{\text{Substrate}}$ and E_{LiO_2} stand for the total energies of the clean substrate and the isolated LiO₂, respectively. As for Co nanoparticle, we choose three representative crystallographic planes ((111), (200) and (220) planes) of Co nanoparticle with face-centred cubic structure which could bind to N-C nanosheet and establish the Co-NPs/N-C catalyst. All layers of the carbon nanosheets are relaxed during the structural optimization.

[S1] G. Kresse, D. Joubert, *Phys. Rev. B* **1999**, 591, 758.

[S2] P.E. Blochl, *Phys. Rev. B* **50** **1994**, 50, 17953.

[S3] J.P. Perdew, K. Burke, M. Ernzerhof, *Phys. Rev. Lett.* **1996**, 77, 3865.

ii) We have adjusted the binding site between Co nanoparticle and N-C substrate and re-optimized the structure of intermediate adsorbate adsorbed on (200) plane of Co nanoparticle, as demonstrated in Supplementary Fig. 37a. The corresponding adsorption energy of LiO₂ for (200) plane of Co nanoparticle is -5.09 eV.

We have corrected the corresponding description in the revised manuscript and supporting information.

Line 21 in page 8:

The absorption structures of LiO₂ on (111), (200) and (220) planes of Co nanoparticle are demonstrated in Fig. 5c and Supplementary Fig. 37, delivering ΔE_{ads} values of -2.75, -5.09 and -3.31 eV, respectively.

Page 22 in the supporting information:

Supplementary Figure 37. a, b) The optimized structures of LiO₂ adsorbed on (200) and (220) planes of Co nanoparticle in Co-NPs/N-C.

iii) According to DFT calculations, much higher adsorption capability between LiO_2 species and Co-SAs/N-C catalyst can be achieved. First, the strong binding interactions can enhance the accessibility of catalytic active sites and boost active species utilization rate efficiently during ORR process. Second, abundant Co- N_4 active centers with higher LiO_2 adsorption capability can function as preferable nucleation sites for the continuous Li_2O_2 growth, which is consistent with those reported works.^[R1,R2] With the reaction further progressing, Co- N_4 configuration cannot adsorb the LiO_2 and act as the ORR sites anymore because their surfaces have already precipitated by the Li_2O_2 . This means the catalytic sites have been passivated, so the Li_2O_2 in this system cannot grow into larger agglomerates. Li_2O_2 passivation limits the accumulation on the cathode surface below ten nanometers. Herein, such large binding energy derived from the desirable isolated metal atoms plays a critical role in forming the special Li_2O_2 morphology with nano size and homogeneous distribution for Co-SAs/N-C catalyst. This is thoroughly different from Co-SAs/N-C and N-C, which demonstrate limited adsorption capability. Fortunately, when recharged in turn, the achievable Li_2O_2 species can make full use of the intimated contacted catalytic sites, which is beneficial for easy decomposition. So the ever-passivated active sites can thoroughly recover their catalytic activity. Similarly, K. R. Yoon et al reported the much strong adsorption energy of LiO_2 (ranging from -3.613 to -8.145 eV) on the Co_4N nanorods also promoted the reversible formation/decomposition of Li_2O_2 .^[R3] G. Cui and L. Chen's group demonstrated that CoO played a key role in improving the cycling stability due to the much higher adsorption energy of LiO_2 for each facet (ranging from -2.5 to -13.4 eV).^[R4] Accordingly, the large adsorption energy of LiO_2 derived from the Co- N_4 configurations may not induce the poisoning or deactivation of the active sites, but just temporary passivation, which would be easily recovered by reversible decomposition of discharge products during the following OER process.

[R1] P. Zhang, R. Wang, M. He, J. Lang, S. Xu, X. Yan, *Adv. Funct. Mater.* **2016**, 26, 1354.

[R2] W. Yao, Y. Yuan, G. Tan, C. Liu, M. Cheng, V. Yurkiv, X. Bi, F. Long, C. R. Friedrich, F. Mashayek, K. Amine, J. Lu, R. S. Yassar, *J. Am. Chem. Soc.* **2019**, 141, 12832.

[R3] K. R. Yoon, K. Shin, J. Park, S. H. Cho, C. Kim, J. W. Jung, J. Y. Cheong, H. R. Byon, H. M. Lee, I. D. Kim, *ACS Nano* **2018**, 12, 128.

[R4] C. Shang, S. Dong, P. Hu, J. Guan, D. Xiao, X. Chen, L. Zhang, L. Gu, G. Cui, L. Chen, *Sci. Rep.* **2015**, 5, 8335.

iv) Following the reviewer's kind suggestion, the calculated accuracy of binding energy has been revised to two digits after the decimal point.

Line 16 in page 8:

The Li atom of LiO_2 is effectively coordinated with N atom, one O atom of LiO_2 is strongly bound with neighboring single Co atomic site. Li and O atoms are simultaneously involved with bonding with Co- N_4 configuration, displaying an intrinsic adsorption energy (ΔE_{ads}) of LiO_2 for Co- N_4 moiety as high as -8.97 eV. In sharp contrast, as depicted in Fig. 5b, without the Co- N_4 configuration, only Li atom of LiO_2 is captured by exposed N atoms, leading to a much lower ΔE_{ads} value of -0.82 eV. The absorption structures of LiO_2 on (111), (200) and (220) planes of Co nanoparticle are demonstrated in Fig. 5c and **Supplementary Fig. 37**, delivering ΔE_{ads} values of -2.75, -5.09 and -3.31 eV, respectively.

REVIEWERS' COMMENTS:

Reviewer #1 (Remarks to the Author):

The authors have addressed my comments and improved the work. I recommend for publication.

Reviewer #2 (Remarks to the Author):

I recommend this paper to be published at current format.